# HOXB9 Overexpression Confers Chemoresistance to Ovarian Cancer Cells by Inducing ERCC-1, MRP-2, and XIAP

**DOI:** 10.3390/ijms24021249

**Published:** 2023-01-08

**Authors:** Dong Hoon Suh, Wook Ha Park, Miseon Kim, Kidong Kim, Jae Hong No, Yong Beom Kim

**Affiliations:** 1Department of Obstetrics and Gynecology, Seoul National University Bundang Hospital, 82 Gumi-ro, 173 Beon-gil, Bundang-gu, Seongnam 13620, Republic of Korea; 2Department of Obstetrics and Gynecology, Seoul National University College of Medicine, 103 Jongno-gu, Seoul 03080, Republic of Korea

**Keywords:** HOXB9, ovarian cancer, apoptosis, chemoresistance, mechanism

## Abstract

The purpose of this study was to identify the role of HOXB9 and associated molecular mechanism in acquiring chemoresistance to ovarian cancer cells. After establishing HOXB9-overexpressing cells (HOXB9-OE/SKOV3), cisplatin resistance-induced cells (Cis-R/SKOV3), and an ovarian cancer xenograft mouse model, the effects of HOXB9 were evaluated in vitro and in vivo. Expression levels of ERCC-1, MRP-2, XIAP, and Bax/Bcl-2 were assessed as putative mechanisms mediating chemoresistance. Cisplatin-induced apoptosis was significantly decreased in HOXB9-OE/SKOV3 compared to SKOV3. Cisplatin treatment of SKOV3 strongly induced ERCC-1, MRP-2, and XIAP, and apoptosis was strongly induced through the inhibition of Bcl-2 and activation of Bax. ERCC-1, MRP-2, XIAP, and Bcl-2 were also strongly induced in HOXB9 OE/SKOV3. In contrast to SKOV3, cisplatin treatment alone of HOXB9 OE/SKOV3 did not affect the expression of Bcl-2 and Bax, and consequently, there was no increase in apoptosis. HOXB9 knockdown suppressed the expression of ERCC-1 and XIAP, but did not affect MRP-2 and Bcl-2/Bax expression in HOXB9 OE/SKOV3 and Cis-R/SKOV3, and caused a small increase in apoptosis. Treatment of SKOV3 with both cisplatin and siRNA_HOXB9 led to complete suppression of ERCC-1, MRP-2, and XIAP, and significantly increased apoptosis through inhibition of Bcl-2 expression and activation of Bax. The results observed in Cis-R/SKOV3 were similar to that in HOXB9 OE/SKOV3. Our data suggest that HOXB9 overexpression may cause chemoresistance in ovarian cancer cells by differential induction of ERCC-1, MRP-2, and XIAP depending on the strength of HOXB9 expression through inhibition of the mitochondrial pathway of apoptosis, including Bax/Bcl-2.

## 1. Introduction

Ovarian cancer is the most lethal gynecologic cancer and has a high risk of relapse [1]. Chemoresistance is the main cause of disease recurrence and metastasis. The majority of the patients receive multiple lines of chemotherapy because of recurrence, but ultimately develop chemo-resistant disease [2]. Once chemoresistance develops, the response rate to subsequent chemotherapy drops to less than 15% with progression-free survival of 3–4 months and median overall survival of less than 1 year [2]. Therefore, it is essential to understand the molecular mechanisms of chemoresistance to develop effective therapeutic strategies for ovarian cancer.

A large body of literature exists on the putative mechanisms of chemoresistance in ovarian cancer. The cytotoxic effect of platinum is associated with the formation of platinum-DNA adducts, which lead to intra- and inter-strand crosslinks, and subsequently, to DNA breaks, as well as mitochondrial damage, which causes decreased ATPase activity and eventually cell death [3]. Therefore, resistance to platinum may result not only from alterations in the transporter proteins, and consequently, reduced uptake of platinum into cells or increased efflux, but also from increased DNA repair following adduct formation, for example, through alterations in various proteins associated with DNA repair mechanisms, including upregulation of excision repair cross-complementation Group 1 (ERCC-1) protein [4], and downregulation of MLH1, MSH2, and MSH1 [5]. Plausible mechanisms also include activation of oncogenes, which modulate the expression of apoptosis-related genes, epithelial-mesenchymal transition, mitochondrial alteration, autophagy, cancer stemness, and inhibition of tumor suppressor genes [6,7,8,9]. Although various genes or signaling pathways are associated with chemoresistance, the molecular mechanism of chemoresistance in ovarian cancer remains unclear and there has been no significant improvement in the survival rate of ovarian cancer patients in the past few decades [10].

Homeobox (HOX) genes, which share a highly conserved 183-base pair DNA sequence, encode transcription factors that regulate not only embryogenic development but also several processes in the adult tissue, including cell proliferation, differentiation, angiogenesis, migration, and apoptosis [11,12,13]. The 39 human HOX genes are located in four clusters A, B, C, and D, each of which contains 9–11 paralogous HOX genes. It is reported that HOX genes are multifunctional and expressed in various solid tumors, including lung, thyroid, prostate, breast, and ovarian cancer [13]. Although the function of HOX genes in ovarian cancer remains unclear, HOXB9 is known to have multiple roles in tumor growth, angiogenesis, chemoresistance, as well as immune evasion as a suppressor of activated human T cells in cancer [14,15,16,17]. Our previous study found that HOXB9 was overexpressed in RMUG-S, a platinum-resistant mucinous ovarian cancer cell line, but not in SKOV3, and inhibition of HOXB9 in RMUG-S rendered the cells platinum-susceptible and effectively induced apoptosis [18].

In this study, we investigated the function of HOXB9 in the chemoresistance of ovarian cancer cells and its molecular mechanism using HOXB9-overexpressing SKOV3 cells and a mouse xenograft model.

## 2. Results

### 2.1. HOXB9 Overexpression Induced Cell Growth, Proliferation and Cancer Signaling Pathway

To identify the effect of HOXB9 overexpression in ovarian cancer progression, the high-grade serous ovarian cancer cell line SKOV3 was selected. Overexpressed plasmids of HOXB9 were used to transfect SKOV3 cells and induce HOXB9-overexpression (HOXB9 OE/SKOV3). Cell viability and proliferation were checked for 1–5 days using Presto Blue and CCK-8 assay, respectively, to evaluate the effect of HOXB9-overexpression on cell growth in SKOV3. Cell viability was higher in HOXB9 OE/SKOV3 than SKOV3 and the differences between the two cell lines were statistically significant on Day 3 and thereafter (Figure 1A). Cell proliferation was also higher in HOXB9 OE/SKOV3 than SKOV3 (Figure 1B). Therefore, overexpression of HOXB9 promotes cell growth in SKOV3. Additionally, to investigate whether HOXB9-overexpression was related to cancer progression, we determined the expression level of genes related to EMT, angiogenesis, cancer stemness and apoptosis using RT-PCR. As shown in Figure 1C, mRNA levels of N-Cadherin, Vimentin, VEGF (vascular endothelial growth factor), IGF (insulin-like growth factor), Nanog, Oct-4 (octamer-binding transcription factor 4), Sox2 (sex-determining region Y box transcription factor-2), and Bcl-2 (B-cell lymphoma-2) were higher in HOXB9 OE/SKOV3 than SKOV3. However, the mRNA level of E-cadherin, Bax (Bcl-2-associated X protein) was lower in HOXB9 OE/SKOV3 than SKOV3. These results demonstrated that the overexpression of HOXB9 promoted nucleus accumulation and the expression of tumor-progression-related genes.

### 2.2. Overexpressed HOXB9 Inhibited Cisplatin-Induced Apoptotic Cell Death

To explore the correlation between HOXB9 and cisplatin resistance in ovarian cancer cells, we performed a Presto Blue assay to evaluate the effect of HOXB9 overexpression on cisplatin-induced apoptotic cell death. The cell viability and proliferation of SKOV3 decreased in a dose-dependent manner (cisplatin concentration, 0, 10, 50, 100 µM). However, cisplatin caused lower cell death in HOXB9 OE/SKOV3 than in SKOV3 and the dose-dependent decrease in cell viability caused by cisplatin treatment was significantly reduced in HOXB9 OE/SKOV3 compared to SKOV3 (Figure 2A, B). Western blot analysis showed that HOXB9 overexpression was associated with increased expression of the anti-apoptotic protein Bcl-2 and decreased expression of cleaved caspase-3 and cleaved PARP, following cisplatin treatment in a dose-independent manner (10, 50, and 100 μM) (Figure 2C). Flow cytometric analysis of Annexin V-FITC and PI staining showed that apoptosis was increased in SKOV3, following cisplatin treatment in a dose-dependent manner, but not in HOXB9 OE/SKOV3 (Figure 2D). Quantitative analysis demonstrated that apoptosis was significantly decreased in HOXB9 OE/SKOV3 than in SKOV3 for each concentration of cisplatin tested. A caspase-3 activity assay revealed that cisplatin treatment induced caspase-3 activity in a dose-dependent manner in SKOV3 cells, but not in HOXB9 OE/SKOV3 cells (Figure 2E). These results demonstrated that HOXB9 overexpression may suppress cisplatin-induced apoptosis in vitro.

### 2.3. HOXB9 Overexpression Promoted the Tumor Progression in Mouse Xenograft Model

To further investigate whether the HOXB9 overexpression induced tumorigenesis, we established a mouse xenograft model using the SKOV3 or HOXB9 OE/SKOV3 cell. We found that the growth rate of tumors in the HOXB9-overexpressing group was significantly faster than that in the control SKOV3 group (Figure 3A). At the time of sacrifice, the tumors were resected from the xenograft mice (Figure 3B,C). The tumors from the control group were smaller (944.8 mm^3^ ± 454.9 mm^3^ vs. 2159.4 mm^3^ ± 75.4 mm^3^; *p* < 0.01) and weighed less (640.0 mg ± 239.4 mg vs. 1186.0 mg ± 473.2 mg; *p* = 0.035) than those from the HOXB9-overexpressing group. Immunoblotting analysis data demonstrated that the protein expression levels of HOXB9, Bcl-2, and Survivin were significantly higher in the tumors from the HOXB9-overexpressing group than from the control group (Figure 3D). Immunohistochemical staining analysis of tumor tissue showed that HOXB9 was strongly expressed in the tumors of HOXB9 OE/SKOV3-injected group compared with those of SKOV3-injected group (Figure 3E).

### 2.4. HOXB9 Overexpression Induced Chemoresistance in the Tumor of Mouse Xenograft Model

Mice were injected with cisplatin 10 mg/kg (intraperitoneal) every 2 days. The tumor growth of SKOV3-injected group was inhibited by cisplatin treatment. The tumor growth began to grossly diminish 15 days after cisplatin treatment. Thereafter, tumor volume continued to decrease through the monitoring period (Figure 4A). On the contrary, the tumor of HOXB9 OE/SKOV3-injected group kept growing despite cisplatin treatment which led just to a slope-down of growth rate of the tumor. The HOXB9 OE/SKOV3-injected group was heavier than the SKOV3-injected group after four weeks of cancer cell injection before cisplatin treatment (Figure 4B, 23.1 mg ± 0.9 mg vs. 21.3 mg ± 0.7 mg; *p* = 0.004). Cisplatin effect on body weights of both mice groups was minimal, but seemed slightly prominent at the HOXB9 OE/SKOV3-injected group without statistical significance.

The difference of the cisplatin effect on tumor progression between the two mouse groups was found in the experiments of isolated tumor evaluation (Figure 4C, D). Isolated tumors from the SKOV3-injected group were significantly reduced in both weight (Figure 4C, 640.0 mg ± 239.4 mg vs. 276.7 mg ± 94.4 mg; *p* = 0.032) and volume (Figure 4D, 944.8 mm^3^ ± 454.9 mm^3^ vs. 125.8 mm^3^ ± 24.3 mm^3^; *p* = 0.004) after cisplatin treatment. In the tumors of the HOXB9 OE/SKOV3-injected group, however, the inhibiting effects of cisplatin on tumor progression were not significant (tumor weight, 1186.0 mg ± 473.2 mg vs. 1156.7 mg ± 401.0 mg; tumor volume, 2159.4 mm^3^ ± 75.4 mm^3^ vs. 1807.1 mm^3^ ± 183.3 mm^3^; *p* > 0.05 for both).

### 2.5. HOXB9 Overexpression Induced the Chemoresistance-Related Markers In Vivo

Western blot analysis showed that protein expression levels of ERCC-1, MRP-2, and XIAP were significantly higher in tumors from the HOXB9 OE/SKOV3-injected group than those from the control SKOV3-injected group (Figure 5A). Immunohistochemical staining of the tumor tissue demonstrated strong expression of ERCC-1, MRP-2, and XIAP in most of the HOXB9 OE/SKOV3-injected group tumors compared with the control SKOV3-injected group tumors (Figure 5B).

### 2.6. Platinum Resistance Induced by ERCC-1, MRP-2, and XIAP Depends on HOXB9 Expression Level

In Figure 6A, B, cell viabilities of SKOV3 cells were significantly decreased after cisplatin treatment in a dose-dependent manner (0, 10, 50, 100 μM; Figure 6A, without siRNA; Figure 6B, with siRNA). Cis-R/SKOV3 showed reduced cytotoxic responses at different doses of cisplatin treatment similar to that in HOXB9 OE/SKOV3 (Figure 6A). The reduced cytotoxic responses of both HOXB9 OE/SKOV3 and Cis-R/SKOV3 compared with those of SKOV3 were statistically significant for every dose of cisplatin treatment (10, 50, 100 μM; Figure 6A). However, the reduced cytotoxic responses to cisplatin (platinum resistance) of the two cell lines were rescued to the level of control SKOV3 and exhibited the same cytotoxic effects at every dose of cisplatin treatment when HOXB9 expression was inhibited by treatment with siRNA-HOXB9 (Figure 6B).

Knockdown of HOXB9 expression using siRNA-HOXB9 treatment significantly increased apoptosis and caspase-3 activity in SKOV3 (both, *p* < 0.05), but not in HOXB9 OE/SKOV3 (Figure 6C, D). SKOV3, which expresses a moderate level of HOXB9, showed strong expression of ERCC-1 and Bcl-2, but no expression of MRP-2, XIAP, and Bax, resulting in weak apoptosis (Figure 6E). Cisplatin treatment of SKOV3 strongly induced ERCC-1, MRP-2, and XIAP, and apoptosis was strongly induced through inhibition of Bcl-2 and activation of Bax. Treatment of SKOV3 with both cisplatin and siRNA-HOXB9 led to complete suppression of ERCC-1, MRP-2, and XIAP expression, but similar levels of apoptosis compared with cisplatin treatment alone. ERCC-1, MRP-2, XIAP, and Bcl-2 were strongly induced in HOXB9 OE/SKOV3 and Cis-R/SKOV3. In contrast to SKOV-3, treatment of HOXB9 OE/SKOV3 and Cis-R/SKOV3 with cisplatin only did not affect the expression of Bcl-2 and Bax, and consequently, there was no increase in apoptosis. The siRNA-HOXB9 mediated knockdown of HOXB9 in HOXB9 OE/SKOV3 and Cis-R/SKOV3 suppressed the expression of ERCC-1 and XIAP, but did not affect MRP-2 and Bcl-2/Bax expression, resulting in a small increase in apoptosis compared with no treatment. Treatment of HOXB9 OE/SKOV3 and Cis-R/SKOV3 with siRNA-HOXB9 and cisplatin suppressed ERCC-1, MRP-2 and XIAP expression and rescued apoptosis through the inhibition of Bcl-2 expression and activation of Bax.

## 3. Discussion

We demonstrated that platinum resistance, which is mediated by cisplatin-induced ERCC-1, MRP-2, and XIAP in SKOV3 ovarian cancer cells (with a moderate level of HOXB9 expression), could be overcome by cisplatin treatment alone through the inhibition of Bcl-2 and activation of Bax. However, HOXB9 overexpression induced strong expression of ERCC-1, MRP-2, and XIAP in HOXB9 OE/SKOV3 and Cis-R/SKOV3 and the high level of HOXB9 expression appears to make the cells refractory to cisplatin treatment without affecting Bcl-2 and Bax expression. These findings suggest that platinum resistance in SKOV3 may depend on the strength/level of HOXB9 expression and a mitochondrial apoptotic pathway, including Bcl-2/Bax could be the common effector pathway for apoptosis when HOXB9-related platinum resistance is overcome. The important role of HOXB9 in the development of chemoresistance was further confirmed by the finding that cisplatin-resistance-induced SKOV3 (Cis-R/SKOV3) expressed HOXB9 strongly and showed the same apoptotic responses to cisplatin or siRNA-HOXB9 treatment as HOXB9 OE/SKOV3.

The Cancer Genome Atlas (TCGA) reports four molecular subtypes (on the basis of gene expression) of high-grade serous ovarian cancer: (1) Immunoreactive; (2) Differentiated; (3) Proliferative; (4) Mesenchymal [19]. Among the four subtypes, the mesenchymal subtype is reported to have the worst prognosis and is characterized by strong expression of the HOX genes [19,20]. Another study from the TCGA that analyzed HOX gene data reported that high expression of HOXB9 was significantly associated with suboptimal residual disease after debulking surgery, which is one of the most important and poor prognostic factors in ovarian cancer [21]. Kelly et al. [17] demonstrated that platinum-resistant high-grade serous ovarian cancer cell lines showed upregulation of HOXB genes, especially HOXB4 and HOXB9. Therefore, they suggested that overexpression of HOXB4 and HOXB9 may be linked to the development of platinum resistance. In addition, Kim et al. [18] recently indicated that HOXB9 overexpression appeared to be associated with platinum resistance in mucinous ovarian cancer, which is known for its low response rate to platinum-based chemotherapy. Nonetheless, studies on the mechanism of chemoresistance mediated by HOXB9 overexpression are lacking.

First, we showed that HOXB9 overexpression in SKOV3 was associated with increased cell viability, proliferation, invasion, and possibly other oncogenic characteristics, including activation of cell cycle progression, EMT, angiogenesis, and cancer stemness, most of which were confirmed in vivo in the xenograft mouse model (Figure 3D). HOXB9 overexpression inhibited cisplatin-induced apoptosis and no cisplatin dose-dependent increase in caspase-3 activity was observed in HOXB9 OE/SKOV3. Although there are limited data on the role of HOXB9 in ovarian cancer in the literature, our findings are consistent with the results of the existing studies. Kim et al. [18] found that the inhibition of HOXB9 expression enhanced apoptosis and suppressed EMT markers, such as vimentin, MMP-9, and Oct4. Wu et al. [22] showed that the anti-angiogenic effect of miRNA-192 in ovarian tumor models was mediated through the regulation of HOXB9 and subsequent downregulation of HIF-1α, IL-1β, and ITGA6.

Regarding chemoresistance following cisplatin treatment apoptosis depends on Bcl-2/Bax expression and Bcl-2/Bax expression depends on the strength/level of HOXB9 expression. A moderate level of HOXB9 expression in SKOV3-induced ERCC-1, but not MRP-2 and XIAP, and showed low apoptotic potential [17]. A certain level of platinum resistance would be expected because ERCC-1, MRP-2, and XIAP were significantly induced by cisplatin treatment in SKOV3. However, unlike SKOV3, overcoming platinum resistance was relatively more difficult in HOXB9 OE/SKOV3 and Cis-R/SKOV3. Interestingly, however, knockdown of HOXB9 in HOXB9 OE/SKOV3 and Cis-R/SKOV3, thereby abrogating the expression of HOXB9, did not significantly rescue apoptosis (Figure 6E). Unlike ERCC-1 and XIAP, which were completely suppressed by HOXB9 knockdown alone, MRP-2 expression in HOXB9 OE/SKOV3 was only partly suppressed by HOXB9 knockdown alone. Although MRP-2, a multi-drug transporter that may confer resistance to cisplatin, was not expected to mediate apoptosis in the absence of cisplatin treatment, it may have inhibited apoptosis through Bcl-2/Bax by a direct or indirect mechanism. Combined treatment of HOXB9 knockdown and cisplatin in HOXB9 OE/SKOV3 and Cis-R/SKOV3 could abrogate that mechanism, leading to significant rescue of apoptosis. An increasing number of studies demonstrated that HOXB9 was associated with resistance to chemotherapy in other cancers [15,23]. Chiba et al. showed that HOXB9 knockdown in pancreatic cancer cell lines elevated the sensitivity of the cells to gemcitabine and nab-paclitaxel treatment and the HOXB9 knockdown effect was rescued to the sensitivity level in control cells upon treatment with TGFβ1 recombinant [23]. The association of HOXB9 and cancer progression was also found in breast and lung cancers [15,24]. Hayashida et al. reported that HOXB9 overexpression enriched the tumor microenvironment with angiogenic factors, enabling tumor vascularization and distant metastasis [15]. HOXB9 was also identified as a target gene of WNT/TCF signaling, enhancing the competence of lung adenocarcinoma cells to colonize the bones and brain [24].

The strength of this study is that we first identified the function of HOXB9 in conferring chemoresistance to ovarian cancer cells and its mechanism using HOXB9-overexpressing ovarian cancer cells and a mouse xenograft model. Nevertheless, there is a limitation in our study. Recently raised questions regarding the histological origin of SKOV3 make it difficult to extrapolate the findings from this study [25]. Not including patient samples in this study is another limitation and this could be further investigated in future research.

Ovarian cancer is characterized by its heterogeneity [25]. There are many different histologic subtypes. Specific HOX genes are known to be overexpressed in certain histologic subtypes [26]. HOXB9 overexpression may be a key genomic change in the mucinous carcinoma of the ovary, a histologic subtype where we have an urgent unmet need for intractable platinum resistance. Our study results implicate that the targeted blocking of ERCC-1, MRP-2, XIAP, and, therefore, activating the mitochondrial apoptotic pathway could be feasible therapeutic strategies for effectively alleviating platinum resistance in HOXB9-overexpressing ovarian mucinous carcinoma in clinical practice.

In conclusion, the platinum resistance of SKOV3 ovarian cancer cells may be different in terms of its refractoriness to cisplatin treatment according to the level of HOXB9 overexpression and the subsequently induced expression of ERCC-1, MRP-2, and XIAP, as well as Bcl-2/Bax expression. MRP-2 may have an additional role in inhibiting apoptosis through Bcl-2/Bax in HOXB9-overexpressing SKOV3 cells. Further in vivo intervention studies and clinical trials are necessary to develop individualized therapeutic strategies focusing on this mechanism.

## 4. Materials and Methods

### 4.1. Antibodies and Reagents

The following antibodies were purchased (Appendix A): primary antibodies to homeobox B9 (HOXB9, Thermo Fisher Scientific, Waltham, MA, USA), matrix metalloproteinase-2, -9 (MMP-2, -9, Cell Signaling Technology, Danvers, MA, USA), B-cell lymphoma 2 (BCL-2, abCAM, Cambridge, UK), BCL2-associated X (BAX, abCAM, Cambridge, UK), E-Cadherin, Survivin, Caspase-3, Cleaved Caspase-3, Cleaved Caspase-9, PARP, Cleaved PARP (Cell Signaling Technology, Danvers, MA, USA), excision repair cross-complementation-1 (ERCC-1, abCAM, Cambridge, UK), multidrug resistance-associated protein 2 (MRP-2, abCAM, Cambridge, UK), X-linked inhibitor of apoptosis protein (XIAP, Cell Signaling Technology, Danvers, MA, USA) and α-tubulin (Cell Signaling Technology, Danvers, MA, USA). The anti-rabbit IgG and anti-mouse IgG HRP (Horseradish Peroxidase)-conjugated secondary antibody were purchased from Cell Signaling Technology (Danvers, MA, USA). Cisplatin was purchased from Selleckchem (Houston, TX, USA).

### 4.2. Cell Culture

SKOV3, a human ovarian cancer cell line with serous histology, was obtained from the American Type Culture Collection (ATCC, Rockville, MD, USA). The SKOV3 cells were cultured at the Roswell Park Memorial Institute 1640 (RPMI) media (Cat# LM 011-01, Welgene, Kyungsan, Korea) supplemented with 10% fetal bovine serum (Cat# 16000-044, Invitrogen, Carlsbad, CA, USA) and 1% penicillin-streptomycin (Cat# 15140122, Invitrogen, Carlsbad, CA, USA) in a humidified chamber with 5% CO_2_ at 37 °C.

### 4.3. Plasmid and Transfection

Preparation of HOXB9-overexpressed cell. For the overexpression of the HOXB9 gene in SKOV3, the pcHOXB9 plasmid was constructed based on the pcDNA3.1 backbone (Addgene, Cambridge, MA, USA). Additionally, a 753-bp fragment of human HOXB9 was subcloned in the HindIII and BamHI site of pcDNA3.1. The human HOXB9 full-length gene was amplified using PCR. The PCR primers used for the amplification of human HOXB9 were as follows: forward 5′-AAGCTTATGTCCATTTCTGGGACGCTT-3′ and reverse 5′-GGATCCCTCTTTGCCCTGCTCCTTATT-3′. The orientation and sequence of the inserted DNA were verified by sequencing and restriction enzyme digestion.

Transient transfection was carried out using Attractene (Qiagen, Valencia, CA, USA) according to the manufacturer’s instructions. The pcHOXB9 or its empty vector were transfected into SKOV3 cells, and the transfected cells changed to a selection medium that contained 1.5 mg/mL of G418 for 2–3 weeks. The G418-resistant stable clones were selected for two weeks. The expression level of human HOXB9 was confirmed using PCR. The cells showing overexpression of HOXB9 were stored at −70 °C.

### 4.4. Cell Viability Assay

Cell viability was examined using a PrestoBlue cell viability reagent (Cat# A13262, Invitrogen, Carlsbad, CA, USA) based on the optical density (OD) value, according to the manufacturer’s protocol. The cells were seeded at 1 × 10^5^ cells/well in a 96-well plate and cultured at 37 °C in an incubator. After incubation for 48 h, the treated cells were incubated with 10% PrestoBlue reagent for 30 min. The OD value was determined by the microplate reader at a wavelength of 540 nm. The cell viability of each group was calculated using the GraphPad Prism software (GraphPad Software, Inc., La Jolla, CA, USA). The experiment was repeated 3–5 times, and the data are expressed as the percent of control.

### 4.5. Cell Proliferation Assay

Cell proliferation assay was evaluated by the Cell Counting Kit-8 (CCK-8) Kit (Dojindo, Kumamoto, Japan). Briefly, the SKOV3 or HOXB9 OE/SKOV3 cells in the 96-well microplate were cultured in DMEM were supplemented with 10% FBS and incubated with cisplatin (0–100 μM) for 48 h. The medium was replaced with fresh DMEM containing CCK-8. CCK8, being nonradioactive, allows sensitive colorimetric assays for the determination of the number of viable cells in cell proliferation assays. After 2 h of incubation, the optical density was measured at OD 450.

### 4.6. Generation of Platinum-Resistance Cell

Cisplatin-resistant SKOV3 cell lines were derived from the original parental cell line by continuous exposure to concentrations of cisplatin. Initially, the cells were exposed to IC50 concentration obtained from cell viability assay. These cells were maintained in cisplatin 20 μM containing RPMI 1640 completed media and subcultured upon reaching 70–80% confluency for 6 weeks. At this point the concentration was increased and the above process was repeated. This development period was carried out for about 5–6 months, and the cell lines collected were named Cis-R/SKOV3 cells. Additionally, the vehicle-treated parental cell line was kept in culture during this period as a control cell line.

### 4.7. Immunofluorescence Staining

The cells were seeded at 3 × 10^4^ cells/well onto 15 mm poly-L-lysine coated cover glass in a 24-well plate and incubated at 37 °C for 24 h. The cells were fixed with 4% paraformaldehyde and permeabilized in 0.1% Triton X-100 for 15 min at RT, and subsequently, washed with phosphate-buffered saline (PBS) 3 times for 3 min each. Following this, cells were blocked with 1% bovine serum albumin (Cat# A7030, Sigma Aldrich, St. Louis, MO, USA) for 60 min at room temperature. After the samples were washed with PBS 3 times for 10 min, the samples were incubated with anti-HOXB9 monoclonal antibody (1:500, Cat# 27967, Cell Signaling Technology, Inc., Danvers, MA, USA) overnight at 4 °C. Subsequent to 3 washes for 3 min each with PBS, the cells were incubated with Alexa Fluor Plus 488-conjugated Goat anti-rabbit immunoglobulin G (1:1000, Cat# A32731, Thermo Fisher Scientific, Waltham, MA, USA) for 2 h at room temperature. The mitochondria were stained with MitoTracker Orange (1:5000, Cat# M7511, Thermo Fisher Scientific, Waltham, MA, USA) for 15 min at room temperature. Fluorescence-labeled HOXB9 or mitochondria were observed and photographed under a fluorescent microscope.

### 4.8. siRNA Transfection

siRNAs for human HOXB9 were synthesized from Genolution (Genolution Pharmaceutical Inc, Seoul, Republic of Korea) and siRNA for control (sc-37007) were purchased from Santa Cruz Biotechnology (Dallas, TX, USA), the HOXB9 targeting-sequence sense: 5′-GGAAGCGAGGACAAAGAGAGG-3′. The transience transfection experiment with synthesized and control siRNAs was performed using Lipofectamine RNAi MAX™ (Invitrogen, 13778075) according to the manufacturer’s instructions (Invitrogen, Waltham, MA, USA). For siRNA transfection, 2.5 × 10^5^ cells were plated in 6-well plates and received 2 mL of fresh growth medium prior to transfection. Then, transfected cells were harvested 48 h after incubation in humidified chamber with 5% CO2 at 37 °C and processed further for Western blotting or RNA analysis as appropriate.

### 4.9. Annexin V-FITC/PI Analysis

The apoptotic ratio was detected by the FITC Annexin V Apoptosis Detection Kit (BD Bioscience, San Jose, CA, USA) according to the protocols provided. The cells at a concentration of 5 × 104 cells/well were seeded in a 12-well plate and incubated for 24 h. At the end of the incubation, the cells were treated with the 0, 50, 100 μM concentrations of cisplatin and incubated for 48 h. After the treatment, the supernatant and cell were harvested and transferred into a falcon tube. The cell pellet resuspended in the binding buffer and 5μL of FITC Annexin V and 5 μL PI were added. Cells were incubated for 15 min in the dark and the apoptotic cells were analyzed using a FACS Calibur (BD Bioscience, San Jose, CA, USA).

### 4.10. Caspase 3/7 Activity

The cells were seeded at 1 × 10^5^ cells/well in white-walled 96-well plates for 24 h and treated with cisplatin for 48 h. The treated cells were further incubated with 100 μL of Caspase-Glo 3/7 Reagent (Cat# G8090, Promega, Madison, WI, USA) at room temperature for 30 min. The luminescence of each sample was measured in by the luminometer manufacturer (Molecular Devices, San Jose, CA, USA). All data are expressed as the percent of control.

### 4.11. Cancer Cell-Derived Mouse Xenograft Model

A total of 40 female BALB/c nude mice (8 W, 15–18 g) were purchased from the ORIENT BIO Inc. (Seongnam, Korea) and maintained under standard environmental conditions. The SKOV3 or HOXB9 OE/SKOV3 cells, at a volume of 1 × 10^7^ cells suspended in 100 μL Matrigel, were subcutaneously injected into the right flank region of BALB/c nude mice (*n* = 10/group, four groups: SKOV3-injected (1) with or (2) without cisplatin treatment; HOXB9 OE/SKOV3-injected (3) with or (4) without cisplatin treatment) to establish the mouse xenograft models. Tumor growth was monitored for four weeks after the implantation of the cells until the tumor volume reached 80–120 mm^3^. Tumor volume was measured using digital calipers, calculated using the formula: width × width × length)/2. After excluding 16 mice (4 for each group) of poor tumor growth, 24 mice were then employed in the study. Mice were treated with cisplatin (10 mg/kg in saline, intraperitoneal) or saline once every 2 days, and the tumor volume was measured once every 3 days. After four weeks, the mice (*n* = 6/group, 24 in total) were sacrificed and tumor tissues were dissected out. All procedures were performed in accordance with the guidelines developed by the National Institutes of Health and following the principles for the care and use of laboratory animals, with all efforts being made to minimize pain.

### 4.12. Immunohistochemical Staining

Tumor tissues of cell-derived xenograft mice were deparaffinized, rehydrated, and washed two times in buffer. To reduce nonspecific background staining due to endogenous peroxidase, the slides were incubated in hydrogen peroxide block for 10 min, and washed 4 times in buffer. The primary antibodies were applied and incubated according to the manufacturers’ recommended protocols; HOXB9 (1:200), ERCC-1 (1:200), MRP-2 (1:200) and XIAP (1:200). Additionally, the slides were washed 4 times in buffer. The slides were then applied with primary antibody enhancer, incubated for 20 min at room temperature, and then washed 4 times in buffer. Afterwards, HRP Polymer was applied to the slides, and the slides were incubated for 30 min at room temperature and washed 4 times in buffer. They were then incubated with hematoxylin for chromogen, and washed 4 times in deionized water.

### 4.13. Reverse Transcription Polymerase Chain Reaction

Total cellular RNA was extracted from the cells using the TRIzol reagent (Cat# 15596026, Thermo Fisher Scientific, Waltham, MA, USA), according to the manufacturer’s instructions. cDNA was synthesized from 2 µg of RNA using a reverse transcription kit (Cat# A3500, Promega, Madison, WI, USA), and the PCR was performed using 2× PCR Master mix Solution (Cat# 25027, Intron Biotechnology, Seongnam, Korea). PCR cycles were: pre-treatment at 95 °C for 5 min, 94 °C for 30 s, 60 °C for 30 s, 72 °C for 30 s (30 cycles), then 72 °C for 10 min and a hold at 4 °C. The PCR products were analyzed on a 1% agarose gel and their band intensities were quantified by densitometry. The expression levels of RNA were normalized to that of 18 s rRNA and shown as bar graphs. Primer information for semi-qRT-PCR is provided in Appendix A.

### 4.14. Western Blot Analysis

Cells were lysed by ice-cold cell lysis buffer (Cat# 17081, Intron Biotechnology, Seongnam, Korea) with the protease inhibitor cocktail. Protein concentrations were determined with a BCA assay kit (Cat# 23117, Thermo Fisher Scientific, Waltham, MA, USA), according to the manufacturer’s instructions. Protein was separated via 10% SDS-PAGE, transferred to a PVDF membrane, and blocked with 5% non-fat milk. Membranes were incubated with anti-HOXB9 (Cat# PA5-101087, Thermo Fisher Scientific, Waltham, MA, USA), anti-ERCC-1 (Cat# ab129267), anti-MRP-2 (Cat# ab172630), anti-Bcl-2 (Cat# ab182858, abCAM, Cambridge, UK), anti-XIAP (Cat# 14334), anti-MMP-2 (Cat# 40994), anti-Bax (Cat# 5023), anti-cleaved caspase-3 (Cat# 9661), anti-cleaved PARP (Cat# 5625, Cell Signaling Technology, Inc., Danvers, MA, USA), and alpha-tubulin (Sigma-Aldrich, St. Louis, MO, USA) antibodies. Next, the membranes were incubated with HRP-conjugated anti-secondary Mouse IgG antibody (Cat# 7076) or HRP-conjugated anti-secondary Rabbit IgG antibody (Cat# 7074, Cell Signaling Technology, Inc., Danvers, MA, USA), and the visualized using Immobilin Forte Western HRP Substrate (Cat# WBLUF0100, Merk Millipore, Burlington, MA, USA).

### 4.15. Statistical Analysis

All the experiments were repeated 3 times and the data were presented as the mean ± SD and analyzed by GraphPad Prism 7 (GraphPad Software, La Jolla, CA, USA). One-way ANOVA with the least significant difference post hoc test was performed to determine statistical significance between groups. Statistical significance was evaluated using paired or unpaired Student’s *t*-test and *p* < 0.05 was considered significant.

## Figures and Tables

**Figure 1 ijms-24-01249-f001:**
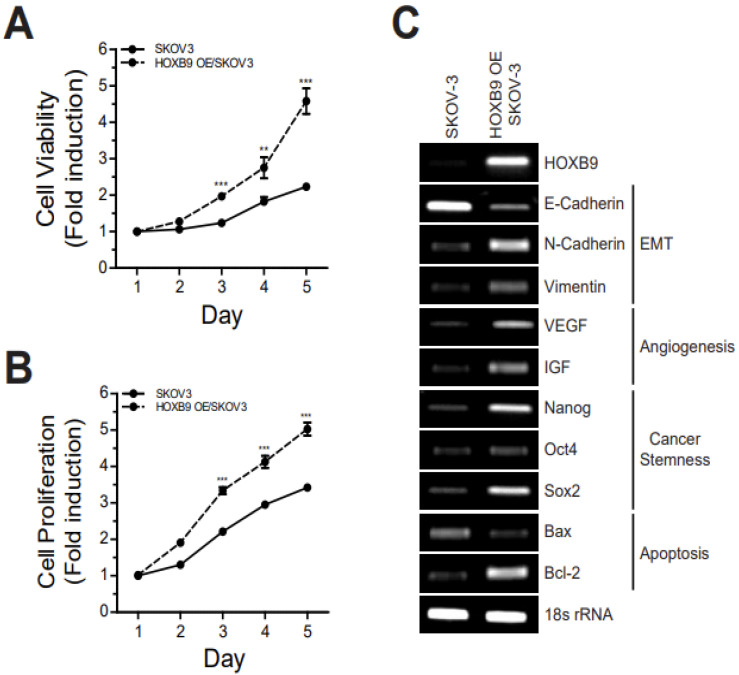
Effect of HOXB9 overexpression in ovarian cancer cells: The SKOV3 and HOXB9 OE/SKOV3 cells were cultured for 1–5 days in 96-well plate, and then, (**A**) cell viability and (**B**) cell proliferation were determined by Presto blue assay and CCK-8 assay, respectively. Each experiment was independently performed in triplicate and data are presented as mean (SD). The two-tailed Student *t* test was used to analyze the differences between Day 1 and 3, 4 and 5. (** *p* < 0.01, *** *p* < 0.001). (**C**) RT-PCR analysis of the EMT (E-Cadherin, N-Cadherin, Vimentin), angiogenesis (VEGF, IGF), cancer stemness (Nanog, Oct4, Sox2) and apoptosis-related gene expression (Bax, Bcl-2) in SKOV3 and HOXB9 OE/SKOV3 cells. Bax, Bcl-2-associated X protein; Bcl-2, B-cell lymphoma 2; EMT, epithelial-mesenchymal transition; IGF, insulin-like growth factor; Oct-4, octamer-binding transcription factor 4; Sox-2, sex-determining region Y box transcription factor-2; VEGF, vascular endothelial growth factor.

**Figure 2 ijms-24-01249-f002:**
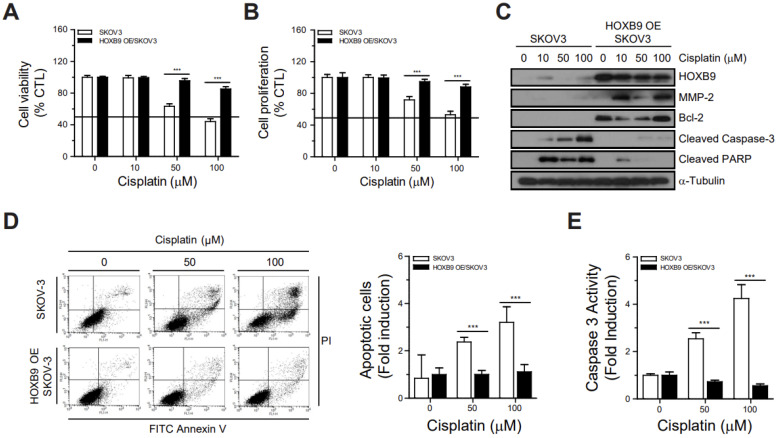
Inhibition of cisplatin-induced apoptotic cell death by HOXB9 overexpression: The SKOV3 and HOXB9 OE/SKOV3 cells were treated with the indicated doses of cisplatin for 48 h, and then, (**A**) cell viability and (**B**) cell proliferation were measured by a Presto blue assay and a CCK-8 assay, respectively; (**C**) Western blot analysis of HOXB9, MMP-2, Bcl-2, cleaved caspase-3, cleaved PARP and α-tubulin in SKOV3 and HOXB9 OE/SKOV3 cells following cisplatin treatment; (**D**) Apoptosis and (**E**) Caspase 3 activity analysis of SKOV3 and HOXB9 OE/SKOV3 cells following cisplatin treatment. Each experiment was independently performed in triplicate: results are expressed as mean ± standard deviation. The two-tailed Student *t* test was used to analyze the differences between the groups. (** *p* < 0.01, *** *p* < 0.001). Bcl-2, B-cell lymphoma-2; MMP-2, matrix metalloproteinase-2.

**Figure 3 ijms-24-01249-f003:**
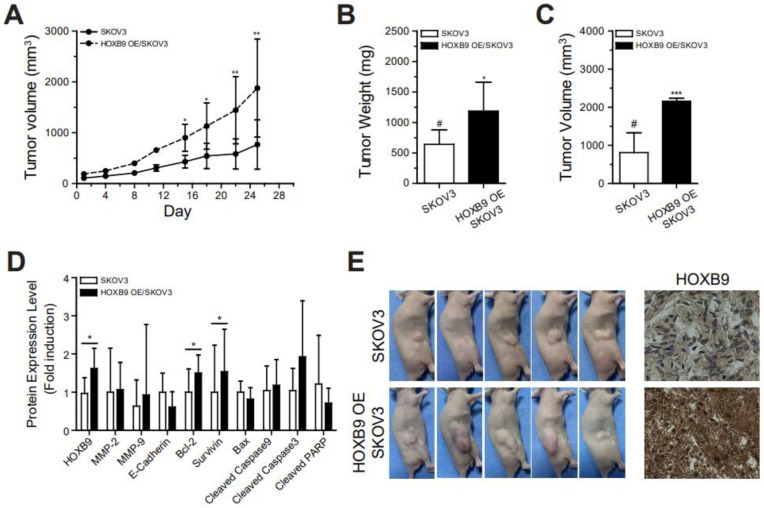
Promotion of tumor growth by HOXB9 overexpression in mouse xenograft models: (**A**) Growth rate of tumors in SKOV3- or HOXB9 OE/SKOV3-injected mouse models. The tumor volume is measured for 4 weeks after the implantation of SKOV3 and HOXB9 OE/SKOV3 cells. The two-tailed Student *t* test was used to analyze the differences between the groups (* *p* < 0.05, ** *p* < 0.01, F value = 3.369, SKOV3-injected vs. HOXB9 OE/SKOV3-injected group, N = 6); (**B**) Mean tumor weight (# control, * *p* < 0.05, F value = 1.080); (**C**) Tumor volume (F value = 1.685) of isolated tumors from mice (# control, *** *p* < 0.001, SKOV3-injected vs. HOXB9 OE/SKOV3-injected group); (**D**) Western blot analysis of HOXB9, MMP-2, MMP-9, E-Cadherin, Bcl-2, Survivin, Bax, Cleaved Caspase-9, Cleaved Caspase-3, and Cleaved PARP protein expression levels in isolated the tumors (SKOV3- or HOXB9 OE/SKOV3-injected tumor) (* *p* < 0.05); (**E**) Gross tumors of the representative mice of each group before harvest and immunohistochemical staining analysis of HOXB9 in the tumors of SKOV3 or HOXB9 OE/SKOV3 cells (magnification: 400×; scale bar = 20 μm). Each experiment was independently performed in triplicate. Bax, BCL2-associated X protein; Bcl-2, B-cell lymphoma 2; MMP-2, matrix metalloproteinase-2; MMP-9, matrix metalloproteinase-9; PARP, Poly (ADP-ribose) polymerase.

**Figure 4 ijms-24-01249-f004:**
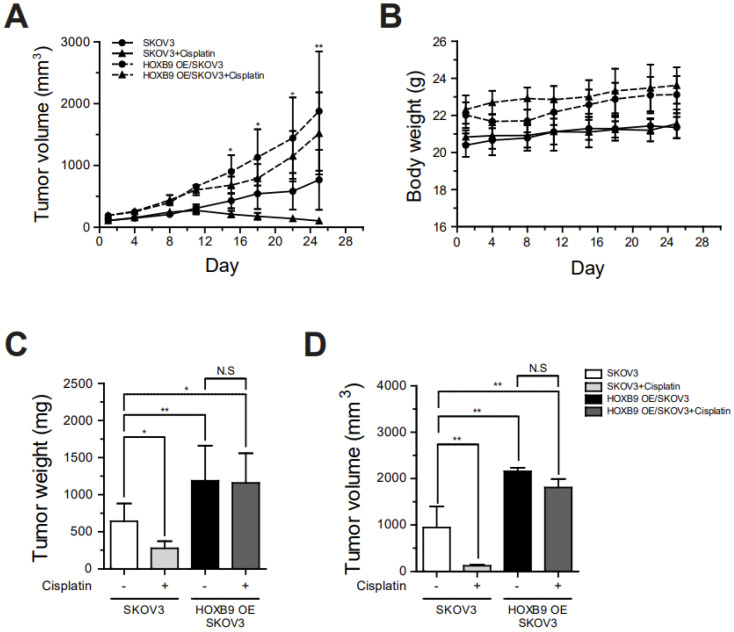
Induction of cisplatin-resistant tumors by HOXB9 overexpression in mouse xenograft models: mice were injected with cisplatin 10 mg/kg (intraperitoneal) every 2 days. (**A**) Growth rate of tumors or (**B**) body weight of mice group with or without cisplatin in SKOV3- and HOXB9 OE/SKOV3-injected mouse model; (**C**) Tumor weight; (**D**) Volume of isolated tumors from mice (SKOV3-injected group vs. SKOV3-injected group with cisplatin, HOXB9 OE/SKOV3-injected group vs. HOXB9 OE/SKOV3-injected group with cisplatin, N = 6). The tumor volume is measured for 4 weeks after the implantation of SKOV3 and HOXB9 OE/SKOV3 cells. Each experiment was independently performed in triplicate. The two-tailed Student *t* test was used to analyze the differences between the groups (* *p* < 0.05, ** *p* < 0.01). N.S: Not Significant.

**Figure 5 ijms-24-01249-f005:**
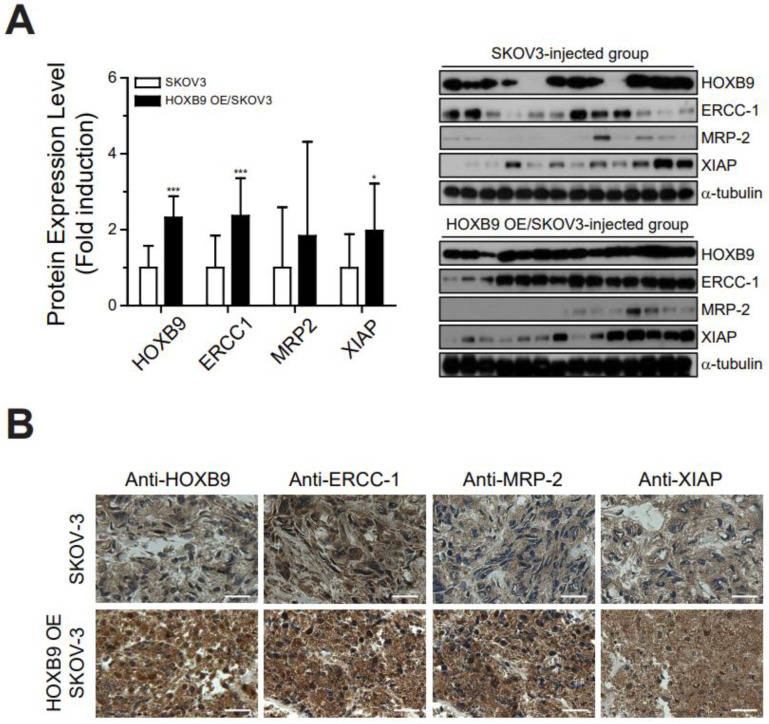
Expression of chemoresistance-related genes in mouse tumors: (**A**) Western blot analysis of HOXB9, ERCC-1, MRP-2 and XIAP in isolated tumors from mice. Each experiment was independently performed in triplicate: results are expressed as mean ± standard deviation. The two-tailed Student *t* test was used to analyze the differences between the groups (* *p* < 0.05, *** *p* < 0.001, SKOV3-injected vs. HOXB9 OE/SKOV3-injected group); (**B**) Immunohistochemical staining analysis of HOXB9, ERCC-1, MRP-2 and XIAP in the tumors of SKOV3 or HOXB9 OE/SKOV3 cells (magnification: 400×; scale bar = 20 μm). Each experiment was independently performed in triplicate. ERCC-1, excision repair cross-complementation-1; MRP-2, multidrug resistance-associated protein-2; XIAP, X-linked inhibitor of apoptosis protein.

**Figure 6 ijms-24-01249-f006:**
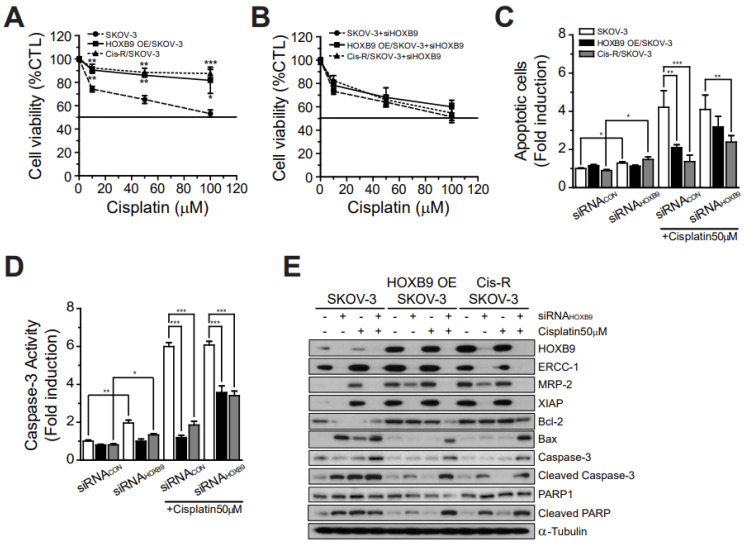
Effect of HOXB9 overexpression on apoptosis and chemoresistance-related genes: cell viability of SKOV3 and HOXB9 OE/SKOV3 cells following cisplatin treatment (**A**) without (F value = 2.603) or (**B**) with siRNA_HOXB9 treatment (F value = 0.05973). The cells were treated with siHOXB9 100 nM or cisplatin 50 μM for 48 h (N = 4); (**C**) Apoptosis (F value = 1.026, N = 4) and (**D**) Caspase 3 activity (F value = 1.619, N = 4) analysis of SKOV3 and HOXB9 OE/SKOV3 cells following cisplatin treatment with or without siRNA_HOXB9 treatment. Each experiment was independently performed in quadruplicate. Results are expressed as mean ± standard deviation. The two-tailed Student *t* test was used to analyze the differences between the SKOV3 and the others group. (**p* < 0.05, ** *p* < 0.001, *** *p* < 0.0001); (**E**) Immunoblot analysis of HOXB9, ERCC-1, MRP-2, XIAP, Bcl-2, Bax, caspase-3, cleaved caspase-3, PARP1, and cleaved PARP in the three cells with or without treatments of siRNA_HOXB9 or cisplatin 50 uM (Appendix A: quantification graphs for the protein expression levels. # Control, **p* < 0.05, ** *p* < 0.001, *** *p* < 0.0001). Immunoblot experiment was independently performed in triplicate. Bax, BCL2 associated X; Bcl2, B-cell lymphoma 2; ERCC-1, excision repair cross-complementation-1; MRP-2, multidrug resistance-associated protein-2; PARP, poly (ADP-ribose) polymerase; XIAP, X-linked inhibitor of apoptosis protein.

## Data Availability

The datasets used and/or analyzed during the current study are available from the corresponding author on reasonable request.

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
