# Peer review of "HOXB9 Overexpression Confers Chemoresistance to Ovarian Cancer Cells by Inducing ERCC-1, MRP-2, and XIAP"

_ijms, 2023, doi:10.3390/ijms24021249_

Round 1
Reviewer 1 Report
Dear Authors
Please find attached a document with my comments for improvement of the manuscript which you may find useful.

Author Response
Response to Reviewer 1 Comments
The study by Suh et al. presents the role of HOXB9 in cisplatin associated chemoresistance in ovarian cancer. The authors through in vitro and in vivo studies showed that HOXB9 overexpression promoted cell proliferation and upregulated EMT, angiogenesis and stemness markers. The authors also showed that HOXB9 overexpression inhibited cisplatin apoptosis whereas a synergistic treatment consisting of HOXB9 knockdown and cisplatin rescued apoptosis and an alteration in the expression of the mitochondrial pathway molecular markers was observed.
The objectives of the study are interesting and well-explored. I suggest some improvements that the authors might find useful.
Thank you so much.
- Introduction
Point1: Line 36: I suggest a more recent reference related to ovarian cancer epidemiology. (e.g CA Cancer J Clin. 2022 Jan;72(1):7-33).
Response 1: Thank you for your comment. We cited recent referecne related to ovarian cancer epidemiology.
Point2: Line 70 & 71: repetition of the phrase “multiple roles”, please rephrase.
Response 2: Thank you for your comment. We rephrased it and avoid the repetition of the phrase, “multiple roles”.
- Results
Point3: Lines 84-102: This paragraph would benefit from English language editing
Response 3: Thank you for your comment. We edited English of the paragraph.
Point4: Line 219: please note typo (HPXB9)
Response 4: Thank you for your comment. We corrected the typo error.
Poinr5: Lines 237-250: Please ensure that each result presented is accompanied by the relevant figure(s) related to each finding described. Additionally, could the authors please clarify the comparison groups that each finding corresponds to? For example: in lines 240-241 the authors found “similar level of apoptosis”, however the comparison groups are advised to be mentioned (e.g control vs siHOXB9, figure 6C, columns 7 vs 10, p>0.05).asd
Response 5: Thank you for your comment. We confirmed that each result presented was accompanied by the relevant figures related to each finding described. And, we made the comparison groups clearer with additionally addressing the control groups.
- Discussion
Point6: Lines 311-328: Are there other studies related to HOXB9 and chemoresistance in ovarian or other types of cancer? It would be interesting for the authors to briefly discuss if data regarding HOXB9-chemoresistance and combined knockdown with chemotherapeutic drugs exist in the literature for ovarian or other types of cancer.
Response 6: Thank you for your comment. We discussed HOXB9 and chemoresistance in ovarian and other cancers (pancreas and breast cancers) with citing a few relevant references.
Point7: Lines 329-334: I suggest the fact that the study did not include patient samples should be mentioned as a limitation and that this could be investigated in future research.
Response 7: Thank you for your comment. Yes, I agree. We mentioned this point as a limitation at discussion section.
Point8: A short paragraph mentioning the implications of this study in clinical practice and what areas the authors suggest should be explored in the context of future research is recommended
Response 8: Thank you for your comment. A paragraph of clinical implication of this study was added by using some parts moved from conclusion.
Point9: The conclusion is advised to be shorter and succinct focusing on the key findings related to hypothesis of the study (HOXB9 and chemoresistance). Most of the information mentioned in the conclusion currently could be used as per comment 3c
Response 9: Thank you for your comment. Moving some parts of the conclusion to the newly constructed paragraph of clinical implication according to your comments made the conclusion shorter and succint.
- Materials and Methods
Point10: Lines 350 & 477 & 502: I suggest that the catalogue number and dilutions for all antibodies used for western blot and immunohistochemistry to be mentioned for both methods. This information could be provided as a supplementary table
Response 10: Thank you for your comment. Catalogue number and dilutions for all antibodies used for western blot and immunohistochemistry were provided in a supplementary table 1.
Point11: Line 433: Could the authors provide more information related to the siRNA transfection? (e.g plate size used, how many cells were seeded, the concentration of siRNA and transfection reagent used)
Response 11: Thank you for your comment. We added more information about siRNA transfection regarding plate size used, how many cells were seeded, the concentration of siRNA and transfection reagent used.
Point12: Line 460: How did the authors decide on the sample size? Was a power calculation performed?
Response 12: Thank you for your comment. We decided the sample size of animal experiments based on relevant references of the similar design and study purpose. It is too small to calculate statistical power.
Point13: Line 474: A statement related to the compliance of the in vivo studies with the ARRIVE guidelines is recommended. The authors could provide the ARRIVE checklist in the supplementary materials.
Response 13: Thank you for your comment. ARRIVE checklist was provided in the supplementary table.
5. Figures:
Point14: The number of biological and technical replicates is advised to be mentioned in all figure legends. Additionally, abbreviation explanation of proteins/genes is advised to be included in every figure
Response 14: Thank you for your comment. The number of biological and technical replicates was mentioned in all figure legends. Abbreviation explanation of proteins/genes was added in every figure.
Point15: If available a picture showing macroscopic comparison of the tumors dissected from the in vivo experiments would be interesting to be added in figure 3
Response 15: Thank you for your comment. Unfortunately, a picture of macroscopic comparison of the tumors harvested from the mice is not available. Instead, we added a picture of gross tumor in situ before harvest in figure 3.
Point16: Figure 6: It would be useful to add a graph indicating the protein expression levels related to the figure 6E.
Response 16: Thank you for your comment. Graphs indicating the protein expression levels related to the figure 6E were added as a supplementary figure 1.
- Supplementary materials
Poinr17: Regarding the original western blot images could the authors add an indication of the samples corresponding to each lane?
Response 17: Thank you for your comment. We added an indication of the samples corresponding to each lane at the original western blot images.
Point18: Does the primer table correspond to the primers used in this study? For instance, I see in figure 1 that some genes such as Vimentin, IGF, N-cadherin were not mentioned in the table whereas others such as p21, p27 are not mentioned in the main manuscript.
Response 18: Thank you for your comment. We are sorry for this mistake. We updated the primer tables.

Reviewer 2 Report
Review comments are attached

Author Response
Response to Reviewer 2 Comments
The present manuscript projects an effort to probe into ovarian cancer at the molecular level and investigate the role of HOXB89 in the acquisition of platinum chemoresistance of ovarian cancer cells (complemented with the introduction of a mouse xenograft model). The establishment of HOXB89 as a key molecule in the signaling pathway, related to apoptosis and thus drug-resistance, was key to further unraveling the molecules associated with its overexpression in SKOV3 cells, thereby providing keen insight into the potential of overcoming platinum resistance in ovarian cancer. The ensuing work into the suppression of ERCC-1, MRP-2, XIAP, and significantly increased apoptosis, through inhibition of Bcl-2 expression and activation of Bax, was very important in pointing out potential clinical protocols, bypassing drug chemoresistance. The work was carried out competently, with the results portraying eloquently the importance of the involved molecules in the associated mechanistic schemes of cancer cell survival under chemoresistance conditions.
There are some minor corrections that apply to phraseological statements in the results section of the manuscript.
Point1: In section 2.4, the statement “Tumor began to grossly diminish days after cisplatin treatment and thereafter the tumor volume continued to decrease through the monitoring period (Figure 4A).” should be corrected to read “Tumor growth began to grossly diminish days after cisplatin treatment. Thereafter, tumor volume continued to decrease through the monitoring period (Figure 4A).”.
Response 1: Thank you for your comment. We revised the statement according to your comments.
Point2: In the same section and the ensuing paragraph, the statement “The difference of cisplatin effect on tumor progression between the two mice groups was found at the experiments of isolated tumor evaluation (Figure 4C, 4D).” should be corrected to read “The difference of cisplatin effect on tumor progression between the two mouse groups was found in the experiments of isolated tumor evaluation (Figure 4C, 4D).”. Based on the aforementioned remarks the manuscript cold be considered following the minor suggested corrections.
Response 2: Thank you for your comment. Yes, we agreed. We revised the statement according to your comments.
